# Clinical Utility of Repeated Urimal Test of Articulation and Phonation for Patients with Childhood Apraxia of Speech

**DOI:** 10.3390/children8121106

**Published:** 2021-12-01

**Authors:** Jung-Hae Yun, So-Min Shin, Su-Min Son

**Affiliations:** Department of Rehabilitation Medicine, College of Medicine, Yeungnam University, Daegu 42415, Korea; junghae5660@gmail.com (J.-H.Y.); nocturne27@naver.com (S.-M.H.)

**Keywords:** childhood apraxia of speech, urimal test, phonation, articulation, coordination, developmental coordination disorder, functional articulation disorder

## Abstract

Childhood apraxia of speech (CAS) causes inconstant oromotor production. We investigated the clinical efficacy of repeated urimal test of articulation and phonation (U-TAP) in CAS patients. Twenty-eight children were recruited: 19 with CAS and 9 with functional articulation disorder (FAD). Four age-matched typically developing children were also recruited. U-TAP was performed twice repeatedly, and the error rate of consonant accuracy (CA) was measured. Preschool Receptive-Expressive Language Scale (PRES) was also performed. The mean U-TAP CA showed a significant difference between the three groups, with 42.04% for CAS, 77.92% for FAD, and 99.68% for the normal group (*p* < 0.05). The mean difference between the two U-TAP CAs was 10.01% for CAS, 0.82% for FAD, and no difference for the normal group, revealing a significant intergroup difference between CAS and FAD (*p* < 0.05). For the expressive and receptive PRES scores, CAS group showed significantly decreased results compared to FAD and normal group. Only in the CAS group, expressive PRES showed significant decrease rather than receptive PRES score. The CAS group showed a significant difference in the two U-TAP CA compared to the FAD and normal groups. This result implies that repeated U-TAP can be useful for supportive diagnostic tool for CAS by detecting poor reliability of phonation.

## 1. Introduction

Among speech sound disorders, childhood apraxia of speech (CAS) is an organic speech disorder characterized by deficits in planning verbal motor function with poor consistency of oromotor movements [1]. In addition to the phonation problem, CAS patients are known to have a high co-occurrence of developmental coordination disorder (DCD), and often have learning problems leading to various developmental problems including cognitive impairment [2,3,4]. CAS patients also have a higher risk of persistent reading and spelling disorder [5,6], which affects their learning abilities such as writing or reading [7,8]. These developmental problems in CAS patients can cause significant depression and anxiety, and they often have a low quality of life and lack self-satisfaction even in adulthood [9,10,11]. Therefore, early diagnosis and treatment of CAS are very important; however, it is challenging to diagnose CAS. The most commonly used diagnostic method is the American Speech Language Hearing Association (ASHA) criteria. Another commonly used diagnostic tool for CAS is Strand’s 10-point checklist [12]. However, these diagnostic methods are not evaluated quantitatively based on clinical characteristics; therefore, the clinical interpretation of the patient may differ depending on the examiners. 

The Urimal Test of Articulation and Phonology (U-TAP) examines articulation and phonology by inducing speech production through reading, naming, and describing situations depicted in pictures. The number of errors and the pattern of errors can be analyzed. U-TAP is widely used in clinical applications because of its ease and simplicity within a relatively short time. In addition to these merits, the U-TAP is known to have a high validity and reliability for testing children’s phonological ability [13,14,15]. A previous study on U-TAP of 14 children with articulation problems and 9 typically developing children aged 4–6 years showed that U-TAP consonant accuracy (CA) was significantly correlated with the ability to form new phonological representations [16]. Another study of patients with articulation problems aged 3–6 years showed a significant relationship between the results of U-TAP and language problems [17]. 

In the current study, we aimed to evaluate the difference in U-TAP consonant accuracy (CA) by performing U-TAP twice repeatedly in age-matched patients with CAS, functional articulation disorder (FAD), and normal groups, and to identify the possibility of U-TAP as a supportive diagnostic tool for CAS patients. 

## 2. Materials and Methods

### 2.1. Subjects

Twenty-eight children who visited a university hospital for language problems were recruited prospectively according to the following inclusion criteria: (1) subjects born full term, with no specific perinatal history; (2) no specific structural abnormality on conventional brain MRI; (3) no specific history of brain trauma or brain surgery; (4) absence of any diagnosed genetic syndrome or neuromuscular disorder, including muscular dystrophy; (5) absence of diagnosed seizure disorder; and (6) absence of diagnosed developmental delay problems including cerebral palsy or intellectual disability other than language problems. Subjects who underwent tongue tie operation due to a short tongue platoon or with any other structural abnormality of the speech organ were excluded from this study. Four age-matched, typically developing children were prospectively recruited for the normal control group. All subjects were examined by a pediatric neurologist for issues related to the inclusion criteria.

The diagnosis of childhood apraxia of speech (CAS) was performed, satisfying three consensus-based clinical features listed in the American Speech and Hearing Association (ASHA) report (2007): (a) inconsistent production of error on consonants and vowels in repeated syllables or word production, (b) lengthening and disruption of co-articulatory transitions between sounds and syllables, and (c) inappropriate prosody, especially in the realization of lexical or phrasal stress. In addition, Strand’s 10-point checklist, which includes 10 segmental and suprasegmental features, was also used for diagnosis. Using Strand’s 10-point checklist, children satisfying at least four of the 10 features within the checklist were diagnosed with CAS [12]. FAD was diagnosed when articulation or phonologic disorder related to linguistic aspects was present with the exclusion of any other organic speech sound disorder based on the average CA criteria for each age of U-TAP [18]. Two speech language pathologists who were blinded to each other’s diagnosis results independently examined all patients. Cases with different diagnoses between the two speech language pathologists were excluded. Age-matched typically developing children were also examined by two speech language pathologists and were diagnosed as normal children. Parents of all subjects included in this study volunteered for this study and gave their written informed consent. The study was reviewed and approved by the institutional review board of our hospital. 

### 2.2. Assessment

All subjects performed the U-TAP twice repeatedly. CA was measured by dividing the number of correctly pronounced consonants by the total number of pronounced consonants [13]. The mean value and the difference between two repeated U-TAP CAs were calculated. Preschool Receptive-Expressive Language Scale (PRES), which tests semantic language and pragmatic language ability, was also performed and the percentile for expressive PRES and receptive PRES scores within one’s age was obtained [19]. U-TAP and PRES were performed consecutively by the one speech language pathologist on the same day.

### 2.3. Statistical Analysis

Data are displayed as mean ± standard deviation (SD). Data were analyzed using the Statistical Package for Social Sciences (SPSS) version 23.0 (IBM Corp., Armonk, NY, USA). Student’s unpaired t test was used to evaluate the differences in the demographic data of age between groups, and the chi-square test was used for comparison of sex demographic data between groups. For comparison of the mean value, difference of two U-TAP CA, and calculated percentile for expressive PRES and receptive PRES score between groups, Kruskal-Wallis analysis was used. The level of statistical significance was set at *p* < 0.05. If a significant difference was detected among the three groups, a Mann-Whitney U post hoc test was used to elucidate the significance of differences between groups. By using Bonferroni method, *p*-values of 0.017 were considered statistically significant. For comparison of calculated mean value between expressive PRES and receptive PRES scores, a Mann-Whitney test was used, the statistical significant level was *p* < 0.05.

## 3. Results

A total of 32 subjects (mean age 4.34 ± 0.48; range, 4–5 years; 18 males) were included. The demographic characteristics are depicted in Table 1. Nineteen subjects were clinically diagnosed with CAS (mean age 4.21 ± 0.42; 10 males), nine subjects were diagnosed with FAD (mean age 4.44 ± 0.53; 5 males), and four normal controls (mean age 4.75 ± 0.50, 3 males) were included. No significant intergroup differences were observed in the demographic data. However, the mean percentage of U-TAP CA showed a significant difference between the groups, with 42.04% for CAS, 77.92% for FAD, and 99.68% for the normal group (*p* < 0.05) (Table 2). In the Mann-Whitney U post hoc test, the significant difference of mean values were observed between normal and FAD, between normal and CAS, and between FAD and CAS groups (*p* < 0.017). Mean difference of repeated U-TAP CAs showed a significant difference between normal (0.00 ± 0.00 (%)) and CAS (10.01 ± 2.86 (%), and between FAD (0.82 ± 1.01) and CAS groups, but there was no significant difference between normal and FAD group (*p* < 0.017). The CAS group showed a significantly increased difference between the two U-TAP CAs compared to the FAD and normal groups. Percentiles calculated according to age for expressive PRES and receptive PRES score also showed significant intergroup differences. There were significant intergroup differences between normal and CAS, and between FAD and CAS groups except for between normal and FAD group (*p* > 0.017). The CAS group showed the lowest, 1.37 ± 2.75%ile for expressive PRES and 17.10 ± 17.45%ile for receptive PRES. FAD showed 64.44 ± 22.33%ile and 73.22 ± 24.04%ile for expressive and receptive PRES, respectively. Normal group showed 92.13 ± 8.41%ile and 91.62 ± 10.58%ile for expressive and receptive PRES, respectively (Figure 1). Intragroup comparison of the mean value of expressive PRES and receptive PRES scores showed significantly decrease of expressive PRES score than receptive PRES only in the CAS group (*p* < 0.05) (Table 3).

## 4. Discussion

In the current study, we evaluated the difference in CA using repetitive U-TAPs and showed that CAS patients had a significant difference compared to FAD and normal children. The mean value of CA, expressive PRES and receptive PRES scores showed an increase in the order of CAS, FAD, and normal groups. In contrast, the difference between two repetitive U-TAP CA showed a decrease in the order of CAS, FAD, and the normal group. 

There have been few studies on CAS. In 2017, Jenya et al. reported speech inconsistency as a core feature of CAS [20]. In addition, a study by Betz et al. confirmed the characteristics of CAS as error inconsistency in repeated syllables and word production [21]. Our study showed results consistent with those of previous studies. Unlike the FAD and normal groups, the CAS group showed prominent inconsistency during speech production, with a CA difference of 9.1~17.3% in all patients, despite tests conducted consecutively on the same day. This lack of phonation reliability is consistent with the clinical consensus of CAS patients presented by ASHA or Strand’s 10-point checklist. In contrast, normal children showed no difference between the two U-TAPs, and the FAD group showed an insignificant difference of 0–2.4%. The difference between these two groups was not statistically significant (Table 2). To be more specific, only 4 out of 9 FAD patients showed a difference in CA, while 5 showed no difference. Four FAD patients who showed differences in the two U-TAPs had error differences in only one word. A previous study by Oh et al. analyzed the error patterns of articulation among FAD patients. Their results showed that FAD patients had difficulties in pronouncing notes of high proficiency age, which had a specific pattern of being substituted or distorted by the notes of low proficiency age [22]. Their result is consistent with our results, in which FAD patients did not show a prominent difference in two successive U-TAPs. They showed a similar error pattern caused by a simple error due to the delay in becoming proficient in motor production or an error due to the articulation habit among early age children. Moreover, the result of no difference in the normal control group was also consistent with the aforementioned results.

Interestingly, contrary to the difference in CA, the mean value of the CAs showed a prominent increase in the order of CAS, FAD, and the normal group, showing 42.04%, 77.92%, and 99.68%, respectively. This result indicates that CAS patients usually have decreased phonation and articulation ability compared to the FAD or normal control group. However, some patients with CAS showed higher CA than age-matched FAD patients in our study. This implies that CAS patients usually have poor phonation consistency, but not necessarily poor phonation accuracy. 

Our results regarding expressive and receptive PRES scores also tended to increase in the order of CAS, FAD, and the normal control group, revealing the lowest language skills in the CAS group. This is consistent with previous studies, which showed that CAS patients had impaired cognitive language functions. Moreover, our results regarding the mean values of PRES and U-TAP showed similar patterns between the three groups (Figure 1). In 2019, Choi et al. studied the correlation between the values of articulation tests and language tests. Thirty-three children aged 3–6 years were enrolled, revealing that U-TAP and PRES showed a significant correlation [17]. This is consistent with our results, which is another interesting finding of our study. In addition, there was significant difference between the express and receptive PRES scores only in the CAS group, showing significantly decreased result of expressive score than receptive score. This result imply that FAD patients can express what they want to speak even if they have articulation problem, but CAS patients have a significantly lower level of expression skill than the level of their receptive language.

To the best of our knowledge, this is the first study to identify the possibility of using repeated U-TAPs as a supportive diagnostic tool for CAS. However, this study has several limitations. First, the small and different sample size for each group due to the strict inclusion criteria was the biggest challenge. This study was conducted prospectively, and very strict age-matched subjects, aged only 4 to 5 years in all groups, were enrolled in consideration, as language development is highly affected by age. Two speech language pathologists independently examined all patients, and the subjects were enrolled only when their diagnoses were the same. A pediatric neurologist also examined all subjects to determine their neurological status. Subjects were excluded if their condition did not meet our inclusion criteria. Brain magnetic resonance imaging (MRI) was performed to exclude language problems due to brain lesions. In addition, there is a lack of detailed clinical information on patients, such as the diagnosis of DCD. This was a preliminary study, and it is considered that further study through detailed clinical evaluations with a large number of cases is necessary.

## Figures and Tables

**Figure 1 children-08-01106-f001:**
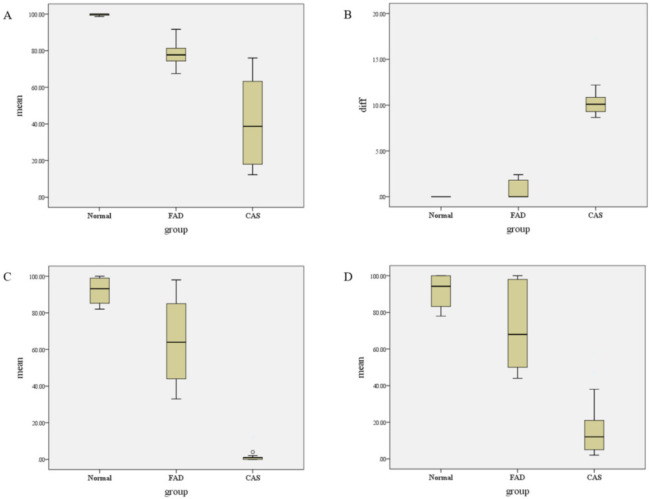
Intergroup comparisons of the mean value and difference of two U-TAP and PRES expressive percentile calculated for age. (**A**): Intergroup comparisons of the mean values of two consecutive U-TAP CAs. (**B**): Intergroup comparisons of differences between two consecutive U-TAP CAs. (**C**): Intergroup comparisons of mean PRES value (E). (**D**): Intergroup comparisons of mean PRES value (R).

**Table 1 children-08-01106-t001:** Demographic data of subjects.

	Age (Years)	Male (%)
Normal group	4.75 ± 0.50	3(75.0)
Functional articulation disorder group (FAD)	4.44 ± 0.53	5(55.6)
Childhood apraxia of speech group (CAS)	4.21 ± 0.42	10(52.6)
*p* value		
†	0.355	0.506
††	0.115	0.412
†††	0.264	0.885

Values are presented as mean ± standard deviation. † between the normal group and FAD group, †† between the normal group and CAS group, ††† between the FAD and CAS groups.

**Table 2 children-08-01106-t002:** Intergroup comparisons of the mean value and difference of two U-TAP CA, PRES.

	Mean (%)	Difference (%)	PRES(E) (%ile)	PRES(R) (%ile)
Normal group	99.68 ± 0.65	0.00 ± 0.00	92.13 ± 8.41	91.62 ± 10.58
FAD group	77.92 ± 7.71	0.82 ± 1.01	64.44 ± 22.33	73.22 ± 24.04
CAS group	42.04 ± 23.83	10.01 ± 2.86	1.37 ± 2.75	17.10 ± 17.45
*p* value	<0.00 *	<0.00 *	<0.00 *	<0.00 *
Mann-Whitney U post hoc test				
*p* value				
†	<0.00 **	0.26	0.03	0.26
††	<0.00 **	<0.00 **	<0.00 **	<0.00 **
†††	< 0.00 **	< 0.00 **	< 0.00 **	< 0.00 **

Values are presented as mean ± standard deviation. Abbreviations: U-TAP, Urimal Test of Articulation and Phonation; CAS, Childhood apraxia of speech group; FAD, functional articulation disorder; PRES (E), Preschool Receptive-Expressive Language Scale (Expressive); PRES (R), Preschool Receptive-Expressive Language Scale (Receptive); CA, Consonant accuracy. † between the normal group and FAD group, †† between the normal group and CAS group, ††† between the FAD and CAS groups * Statistical significance was accepted for *p*-values < 0.05, ** Statistical significance was accepted by using Bonferroni method for *p* < 0.017.

**Table 3 children-08-01106-t003:** Intragroup comparisons of the mean value of PRES (E) and PRES (R).

	PRES(E) (%ile)	PRES(R) (%ile)	*p* value
Normal group	92.13 ± 8.41	91.62 ± 10.58	0.89
FAD group	64.44 ± 22.33	73.22 ± 24.04	0.34
CAS group	1.37 ± 2.75	17.10 ± 17.45	<0.00 *

Values are presented as mean ± standard deviation. Abbreviations: CAS, Childhood apraxia of speech group; FAD, functional articulation disorder; Preschool Receptive-Expressive Language Scale (Expressive); PRES (R), Preschool Receptive-Expressive Language Scale (Receptive). * Statistical significance was accepted for *p*-values < 0.05.

## Data Availability

All data are provided either in the paper.

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
