# Peer review of "Clinical Utility of Repeated Urimal Test of Articulation and Phonation for Patients with Childhood Apraxia of Speech"

_children, 2021, doi:10.3390/children8121106_

Round 1

Reviewer 1 Report

Thank you for your good research.
There are some unclear parts of the this research, so I would like to give you an opinion.

1. Minor
(1) Please clarify the method/

1) It is not recorded in the manuscript that how many CAS and FAD groups are each.
2) When you diagnosis the FAD group, it is necessary to write down what diagnostic methods were used. (U-TAP?)
3) When conducting PRES and U-TAP in CAS, FAD, and Normal groups, we need details such as whether the same examiner performed it or the same day.
4) When conducting U-TAP twice, detailed explanation of whether the same examiner performed it or when the same day or how long duration between two examinations.
(2) Result
In the case of PRES, 77.92% of the values from the FAD group can be viewed as almost a normal range, and please describe how the authors accept the analysis of the values from FAD.

2. Major
1. In statistical analysis, there were 4 in the Normal group and 9 in the FAD group, so this study is less accurate in using parametric statistics. We recommend using nonparametric statistics.

2. When comparing the three groups, it is recommended to use the Kruskal-Wallis & Post-hoc method, which is a nonparametric method of ANOVA, rather than the student unpaired test.

Author Response

Thank you for your good research.
There are some unclear parts of the this research, so I would like to give you an opinion.

1. Minor
(1) Please clarify the method/

1) It is not recorded in the manuscript that how many CAS and FAD groups are each.

Answer: Thank you for your comments. In the abstract and result section, I described the number of each group as follows.

Abstract: Twenty-eight children were recruited: 19 with CAS and 9 with functional articulation disorder (FAD). Four age-matched typically developing children were also recruited.

Results: A total of 32 subjects (mean age 4.34 ± 0.48; range, 4–5 years; 18 males) were included. The demographic characteristics are depicted in Table 1. Nineteen subjects were clinically diagnosed with CAS (mean age 4.21± 0.42; 10 males), nine subjects were diagnosed with FAD (mean age 4.44 ± 0.53; 5 males), and four normal controls (mean age 4.75 ± 0.50, 3 males) were included.

2) When you diagnosis the FAD group, it is necessary to write down what diagnostic methods were used. (U-TAP?)

Answer: I appreciate your comments. Following your comments, we revised the manuscript.

Subjects: FAD was diagnosed when articulation or phonologic disorder related to linguistic aspects was present with the exclusion of any other organic speech sound disorder based on the average CA criteria for each age of U-TAP [18].

3) When conducting PRES and U-TAP in CAS, FAD, and Normal groups, we need details such as whether the same examiner performed it or the same day.

Answer: Thank you for your comment. We revised the manuscript as follows.

Assessment: All subjects performed the U-TAP twice repeatedly. CA was measured by dividing the number of correctly pronounced consonants by the total number of pronounced consonants [13]. The mean value and the difference between two repeated U-TAP CAs were calculated. Preschool Receptive-Expressive Language Scale (PRES), which tests semantic language and pragmatic language ability, was also performed and the percentile for expressive PRES and receptive PRES scores within one’s age was obtained [19,20]. U-TAP and PRES were performed consecutively by the one speech language pathologist on the same day.

4) When conducting U-TAP twice, detailed explanation of whether the same examiner performed it or when the same day or how long duration between two examinations.

Answer: I appreciate your comments. We revised the manuscript as follows.

Assessment: All subjects performed the U-TAP twice repeatedly. CA was measured by dividing the number of correctly pronounced consonants by the total number of pronounced consonants [13]. The mean value and the difference between two repeated U-TAP CAs were calculated. Preschool Receptive-Expressive Language Scale (PRES), which tests semantic language and pragmatic language ability, was also performed and the percentile for expressive PRES and receptive PRES scores within one’s age was obtained [19,20]. U-TAP and PRES were performed consecutively by the one speech language pathologist on the same day.

5) Result
In the case of PRES, 77.92% of the values from the FAD group can be viewed as almost a normal range, and please describe how the authors accept the analysis of the values from FAD.

Answer: The expressive value of FAD group was correctly indicated in the results section of text (66.44 ± 22.33%ile), but the value in the abstract and table (77.92 ± 7.71%ile) was indicated incorrectly. We sincerely apologize for our mistake. We revised the table and figure as follows.

Abstract: Abstract: Childhood apraxia of speech (CAS) causes inconstant oromotor production. We investigated the clinical efficacy of repeated urimal test of articulation and phonation (U-TAP) in CAS patients. Twenty-eight children were recruited: 19 with CAS and 9 with functional articulation disorder (FAD). Four age-matched typically developing children were also recruited. U-TAP was performed twice repeatedly, and the error rate of consonant accuracy (CA) was measured. Preschool Receptive-Expressive Language Scale (PRES) was also performed. The mean U-TAP CA showed a significant difference between the three groups, with 42.04% for CAS, 77.92% for FAD, and 99.68% for the normal group (p<0.05). The mean difference between the two U-TAP CAs was 10.01% for CAS, 0.82% for FAD, and no difference for the normal group, revealing a significant intergroup difference between CAS and FAD (p<0.05). For the expressive and receptive PRES scores, CAS group showed significantly decreased results compared to FAD and normal group. Only in the CAS group, expressive PRES showed significant decrease rather than receptive PRES score. The CAS group showed a significant difference in the two U-TAP CA compared to the FAD and normal groups. This result implies that repeated U-TAP can be useful for supportive diagnostic tool for CAS by detecting poor reliability of phonation.

Table 2. Intergroup comparisons of the mean value and difference of two U-TAP CA, PRES

Mean (%)

Difference (%)

PRES(E) (%ile)

PRES(R) (%ile)

Normal group

99.68 ± 0.65

0.00 ± 0.00

92.13 ± 8.41

91.62 ± 10.58

FAD group

77.92 ± 7.71

0.82 ± 1.01

64.44 ± 22.33

73.22 ± 24.04

CAS group

42.04 ± 23.83

10.01 ± 2.86

1.37 ± 2.75

17.10 ± 17.45

p value

<0.00*

<0.00*

<0.00*

<0.00*

Mann-Whitney U post hoc test

p value

<0.00**

0.26

0.03

0.26

††

<0.00**

<0.00**

<0.00**

<0.00**

†††

< 0.00**

< 0.00**

< 0.00**

< 0.00**

Values are presented as mean ± standard deviation

Abbreviations: U-TAP, Urimal Test of Articulation and Phonation; CAS, Childhood apraxia of speech group; FAD, functional articulation disorder; PRES (E), Preschool Receptive-Expressive Language Scale (Expressive); PRES (R), Preschool Receptive-Expressive Language Scale (Receptive); CA, Consonant accuracy

†between the normal group and FAD group, ††between the normal group and CAS group, †††between the FAD and CAS groups

* Statistical significance was accepted for p-values < 0.05, ** Statistical significance was accepted by using Bonferroni method for p<0.017

Figure 1. Intergroup comparisons of the mean value and difference of two U-TAP and PRES expressive percentile calculated for age

A: Intergroup comparisons of the mean values of two consecutive U-TAP CAs.

B: Intergroup comparisons of differences between two consecutive U-TAP CAs.

C: Intergroup comparisons of mean PRES value (E).

D: Intergroup comparisons of mean PRES value (R).

  1. Major
    1. In statistical analysis, there were 4 in the Normal group and 9 in the FAD group, so this study is less accurate in using parametric statistics. We recommend using nonparametric statistics.

Answer: I appreciate your comments for the manuscript. Following your comments, we performed the Kruskal-Wallis & Post-hoc method and revised the manuscript.

2.3. Statistical analysis

Data are displayed as mean ± standard deviation (SD). Data were analyzed using the Statistical Package for Social Sciences (SPSS) version 23.0 (IBM Corp., Armonk, NY, USA). Student’s unpaired t test was used to evaluate the differences in the demographic data of age between groups, and the chi-square test was used for comparison of sex demographic data between groups. For comparison of the mean value, difference of two U-TAP CA, and calculated percentile for expressive PRES and receptive PRES score between groups, Kruskal-Wallis analysis was used. The level of statistical significance was set at P < 0.05. If a significant difference was detected among the three groups, a Mann-Whitney U post hoc test was used to elucidate the significance of differences between groups. By using Bonferroni method, p-values of 0.017 were considered statitstically significant. For comparison of calculated mean value between expressive PRES and receptive PRES scores, a Mann-Whitney test was used, the statistical significant level was p < 0.05.

  1. Results

A total of 32 subjects (mean age 4.34 ± 0.48; range, 4–5 years; 18 males) were included. The demographic characteristics are depicted in Table 1. Nineteen subjects were clinically diagnosed with CAS (mean age 4.21 ± 0.42; 10 males), nine subjects were diagnosed with FAD (mean age 4.44 ± 0.53; 5 males), and four normal controls (mean age 4.75 ± 0.50, 3 males) were included. No significant intergroup differences were observed in the demographic data. However, the mean percentage of U-TAP CA showed a significant difference between the groups, with 42.04% for CAS, 77.92% for FAD, and 99.68% for the normal group (p < 0.05) (Table 2). In the Mann-Whitney U post hoc test, the significant difference of mean values were observed between normal and FAD. between normal and CAS, and between FAD and CAS groups (p < 0.017). Mean difference of repeated U-TAP CAs showed a significant difference between normal (0.00 ± 0.00 (%)) and CAS (10.01 ± 2.86 (%), and between FAD (0.82 ± 1.01) and CAS groups, but there was no significant difference between normal and FAD group (p < 0.017). The CAS group showed a significantly increased difference between the two U-TAP CAs compared to the FAD and normal groups. Percentiles calculated according to age for expressive PRES and receptive PRES score also showed significant intergroup differences. There were significant intergroup differences between normal and CAS, and between FAD and CAS groups except for between normal and FAD group (p > 0.017). The CAS group showed the lowest, 1.37 ± 2.75%ile for expressive PRES and 17.10 ± 17.45%ile for receptive PRES. FAD showed 64.44 ± 22.33%ile and 73.22 ± 24.04%ile for expressive and receptive PRES, respectively. Normal group showed 92.13 ± 8.41%ile and 91.62 ± 10.58%ile for expressive and receptive PRES, respectively (Figure 1). Intragroup comparison of the mean value of expressive PRES and receptive PRES scores showed significantly decrease of expressive PRES score than receptive PRES only in the CAS group (p < 0.05) (Table 3).

  1. When comparing the three groups, it is recommended to use the Kruskal-Wallis & Post-hoc method, which is a nonparametric method of ANOVA, rather than the student unpaired test.

Answer: I appreciate your comments for the manuscript. Following your comments, we revised the manuscript.

2.3. Statistical analysis

Data are displayed as mean ± standard deviation (SD). Data were analyzed using the Statistical Package for Social Sciences (SPSS) version 23.0 (IBM Corp., Armonk, NY, USA). Student’s unpaired t test was used to evaluate the differences in the demographic data of age between groups, and the chi-square test was used for comparison of sex demographic data between groups. For comparison of the mean value, difference of two U-TAP CA, and calculated percentile for expressive PRES and receptive PRES score between groups, Kruskal-Wallis analysis was used. The level of statistical significance was set at P < 0.05. If a significant difference was detected among the three groups, a Mann-Whitney U post hoc test was used to elucidate the significance of differences between groups. By using Bonferroni method, p-values of 0.017 were considered statitstically significant. For comparison of calculated mean value between expressive PRES and receptive PRES scores, a Mann-Whitney test was used, the statistical significant level was p < 0.05.

  1. Results

A total of 32 subjects (mean age 4.34 ± 0.48; range, 4–5 years; 18 males) were included. The demographic characteristics are depicted in Table 1. Nineteen subjects were clinically diagnosed with CAS (mean age 4.21 ± 0.42; 10 males), nine subjects were diagnosed with FAD (mean age 4.44 ± 0.53; 5 males), and four normal controls (mean age 4.75 ± 0.50, 3 males) were included. No significant intergroup differences were observed in the demographic data. However, the mean percentage of U-TAP CA showed a significant difference between the groups, with 42.04% for CAS, 77.92% for FAD, and 99.68% for the normal group (p < 0.05) (Table 2). In the Mann-Whitney U post hoc test, the significant difference of mean values were observed between normal and FAD. between normal and CAS, and between FAD and CAS groups (p < 0.017). Mean difference of repeated U-TAP CAs showed a significant difference between normal (0.00 ± 0.00 (%)) and CAS (10.01 ± 2.86 (%), and between FAD (0.82 ± 1.01) and CAS groups, but there was no significant difference between normal and FAD group (p < 0.017). The CAS group showed a significantly increased difference between the two U-TAP CAs compared to the FAD and normal groups. Percentiles calculated according to age for expressive PRES and receptive PRES score also showed significant intergroup differences. There were significant intergroup differences between normal and CAS, and between FAD and CAS groups except for between normal and FAD group (p > 0.017). The CAS group showed the lowest, 1.37 ± 2.75%ile for expressive PRES and 17.10 ± 17.45%ile for receptive PRES. FAD showed 64.44 ± 22.33%ile and 73.22 ± 24.04%ile for expressive and receptive PRES, respectively. Normal group showed 92.13 ± 8.41%ile and 91.62 ± 10.58%ile for expressive and receptive PRES, respectively (Figure 1). Intragroup comparison of the mean value of expressive PRES and receptive PRES scores showed significantly decrease of expressive PRES score than receptive PRES only in the CAS group (p < 0.05) (Table 3).

Table 2. Intergroup comparisons of the mean value and difference of two U-TAP CA, PRES

Mean (%)

Difference (%)

PRES(E) (%ile)

PRES(R) (%ile)

Normal group

99.68 ± 0.65

0.00 ± 0.00

92.13 ± 8.41

91.62 ± 10.58

FAD group

77.92 ± 7.71

0.82 ± 1.01

64.44 ± 22.33

73.22 ± 24.04

CAS group

42.04 ± 23.83

10.01 ± 2.86

1.37 ± 2.75

17.10 ± 17.45

p value

<0.00*

<0.00*

<0.00*

<0.00*

Mann-Whitney U post hoc test

p value

<0.00**

0.26

0.03

0.26

††

<0.00**

<0.00**

<0.00**

<0.00**

†††

< 0.00**

< 0.00**

< 0.00**

< 0.00**

Values are presented as mean ± standard deviation

Abbreviations: U-TAP, Urimal Test of Articulation and Phonation; CAS, Childhood apraxia of speech group; FAD, functional articulation disorder; PRES (E), Preschool Receptive-Expressive Language Scale (Expressive); PRES (R), Preschool Receptive-Expressive Language Scale (Receptive); CA, Consonant accuracy

†between the normal group and FAD group, ††between the normal group and CAS group, †††between the FAD and CAS groups

* Statistical significance was accepted for p-values < 0.05, ** Statistical significance was accepted by using Bonferroni method for p<0.017

Reviewer 2 Report

This is an interesting study that reveals that repeated articulation evaluation results can be different in children with CAS. There is not much data on CAS, thank you for conducting this research.

Mental retardation

  • Although this term is still used in the ICD code, as this term has a negative feeling, intellectual disability might be better to use as in DSM-V

Methods

How was the number of participants calculated? The number of participants in the control group is only four and FAD patients were nine. I wonder if you planned the study by calculating the number of people to discriminate the difference between U-TAPs.

Statistics

As in this study, when the number of people is small and the normal distribution is not guaranteed, it is not statistically correct to compare only the average. This needs to be supplemented.

Results

Children with CAS showed a relatively low percentile of expressive language development. If all children had undergone PRES, please share data of receptive language development. You may add one more graph in figure 1.

“successive U-TAPs”

It would be better to clearly convey that it is a repeated test rather than a series of tests.

To the best of our knowledge, this is the first study to identify the possibility of using 218 repeated U-TAPs as a screening tool for CAS.

  • Do you think a repeated articulation test could be a screening test? As you mentioned in the introduction, there’s no definite quantitative tool to diagnose CAS but only we have diagnostic criteria. I guess repeated tests of U-TAP might be used as a supportive diagnostic tool for CAS rather than a screening.

Author Response

This is an interesting study that reveals that repeated articulation evaluation results can be different in children with CAS. There is not much data on CAS, thank you for conducting this research.

1) Mental retardation

Although this term is still used in the ICD code, as this term has a negative feeling, intellectual disability might be better to use as in DSM-V

Answer: Thank you for your comments. we revised the manuscript as follows.

Subejcts: (6) absence of diagnosed developmental delay problems including cerebral palsy or intellecual disability mental retardation other than language problems. 

2) Methods

How was the number of participants calculated? The number of participants in the control group is only four and FAD patients were nine. I wonder if you planned the study by calculating the number of people to discriminate the difference between U-TAPs.

 Answer: I appreciate your comments. This study was a preliminary study and it was not possible to statistically obtain the sample size because there was no similar previous study. So, I added this as a limitation.

Discussion: However, this study has several limitations. First, the small and different sample size for each group due to the strict inclusion criteria was the biggest challenge. This study was conducted prospectively, and very strict age-matched subjects, aged only 4 to 5 years in all groups, were enrolled in consideration, as language development is highly affected by age. Two speech language pathologists independently examined all patients, and the subjects were enrolled only when their diagnoses were the same. A pediatric neurologist also examined all subjects to determine their neurological status. Subjects were excluded if their condition did not meet our inclusion criteria. Brain magnetic resonance imaging (MRI) was performed to exclude language problems due to brain lesions. In addition, there is a lack of detailed clinical information on patients, such as the diagnosis of DCD. This was a preliminary study, and it is considered that further study through detailed clinical evaluations with a large number of cases is necessary.

3) Statistics

As in this study, when the number of people is small and the normal distribution is not guaranteed, it is not statistically correct to compare only the average. This needs to be supplemented.

Answer: I appreciate your comments. Following your comment, we revised statistical analysis from parametric statistics (Student’s unpaired t-test) to nonparametric statistics (Kruskal-Wallis analysis and Mann-Whitney U post hoc test).

2.3. Statistical analysis

Data are displayed as mean ± standard deviation (SD). Data were analyzed using the Statistical Package for Social Sciences (SPSS) version 23.0 (IBM Corp., Armonk, NY, USA). Student’s unpaired t test was used to evaluate the differences in the demographic data of age between groups, and the chi-square test was used for comparison of sex demographic data between groups. For comparison of the mean value, difference of two U-TAP CA, and calculated percentile for expressive PRES and receptive PRES score between groups, Kruskal-Wallis analysis was used. The level of statistical significance was set at p < 0.05. If a significant difference was detected among the three groups, a Mann-Whitney U post hoc test was used to elucidate the significance of differences between groups. By using Bonferroni method, p-values of 0.017 were considered statitstically significant. For comparison of calculated mean value between expressive PRES and receptive PRES scores, a Mann-Whitney test was used, the statistical significant level was p < 0.05.

  1. Results

A total of 32 subjects (mean age 4.34 ± 0.48; range, 4–5 years; 18 males) were included. The demographic characteristics are depicted in Table 1. Nineteen subjects were clinically diagnosed with CAS (mean age 4.21± 0.42; 10 males), nine subjects were diagnosed with FAD (mean age 4.44 ± 0.53; 5 males), and four normal controls (mean age 4.75 ± 0.50, 3 males) were included. No significant intergroup differences were observed in the demographic data. However, the mean percentage of U-TAP CA showed a significant difference between the groups, with 42.04% for CAS, 77.92% for FAD, and 99.68% for the normal group (p < 0.05). (Table 2). In the Mann-Whitney U post hoc test, the significant difference of mean values were observed between normal and FAD. between normal and CAS, and between FAD and CAS groups (p < 0.017). Mean difference of repeated U-TAP CAs showed a significant difference between normal (0.00 ± 0.00 (%)) and CAS (10.01 ± 2.86 (%), and between FAD (0.82 ± 1.01) and CAS groups, but there was no significant difference between normal and FAD group (p < 0.017). The CAS group showed a significantly increased difference between the two U-TAP CAs compared to the FAD and normal groups. Percentiles calculated according to age for expressive PRES and receptive PRES score also showed significant intergroup differences. There were significant intergroup differences between normal and CAS, and between FAD and CAS groups except for between normal and FAD group (p > 0.017). The CAS group showed the lowest, 1.37 ± 2.75%ile for expressive PRES and 17.10 ± 17.45%ile for receptive PRES. FAD showed 64.44 ± 22.33%ile and 73.22 ± 24.04%ile for expressive and receptive PRES, respectively. Normal group showed 92.13 ± 8.41%ile and 91.62 ± 10.58%ile for expressive and receptive PRES, respectively (Figure 1). Intragroup comparison of the mean value of expressive PRES and receptive PRES scores showed significantly decrease of expressive PRES score than receptive PRES only in the CAS group (p < 0.05) (Table 3).

Table 2. Intergroup comparisons of the mean value and difference of two U-TAP CA, PRES

Mean (%)

Difference (%)

PRES(E) (%ile)

PRES(R) (%ile)

Normal group

99.68 ± 0.65

0.00 ± 0.00

92.13 ± 8.41

91.62 ± 10.58

FAD group

77.92 ± 7.71

0.82 ± 1.01

64.44 ± 22.33

73.22 ± 24.04

CAS group

42.04 ± 23.83

10.01 ± 2.86

1.37 ± 2.75

17.10 ± 17.45

p value

<0.00*

<0.00*

<0.00*

<0.00*

Mann-Whitney U post hoc test

p value

<0.00**

0.26

0.03

0.26

††

<0.00**

<0.00**

<0.00**

<0.00**

†††

< 0.00**

< 0.00**

< 0.00**

< 0.00**

Values are presented as mean ± standard deviation

Abbreviations: U-TAP, Urimal Test of Articulation and Phonation; CAS, Childhood apraxia of speech group; FAD, functional articulation disorder; PRES (E), Preschool Receptive-Expressive Language Scale (Expressive); PRES (R), Preschool Receptive-Expressive Language Scale (Receptive); CA, Consonant accuracy

†between the normal group and FAD group, ††between the normal group and CAS group, †††between the FAD and CAS groups

* Statistical significance was accepted for p-values < 0.05, ** Statistical significance was accepted by using Bonferroni method for p<0.017

4) Results

Children with CAS showed a relatively low percentile of expressive language development. If all children had undergone PRES, please share data of receptive language development. You may add one more graph in figure 1.

Answer: I appreciate your opinion. We added the on more graph in figure 1. In addition, a statistical test was performed on intragroup comparison between expressive PRES score and receptive PRES score, and the results are presented in table 3 and described in the manuscript as follows.

Figure 1. Intergroup comparisons of the mean value and difference of two U-TAP and PRES expressive percentile calculated for age

A: Intergroup comparisons of the mean values of two consecutive U-TAP CAs.

B: Intergroup comparisons of differences between two consecutive U-TAP CAs.

C: Intergroup comparisons of mean PRES value (E).

D: Intergroup comparisons of mean PRES value (R).

Table 2. Intergroup comparisons of the mean value and difference of two U-TAP CA, PRES

Mean (%)

Difference (%)

PRES(E) (%ile)

PRES(R) (%ile)

Normal group

99.68 ± 0.65

0.00 ± 0.00

92.13 ± 8.41

91.62 ± 10.58

FAD group

77.92 ± 7.71

0.82 ± 1.01

64.44 ± 22.33

73.22 ± 24.04

CAS group

42.04 ± 23.83

10.01 ± 2.86

1.37 ± 2.75

17.10 ± 17.45

p value

<0.00*

<0.00*

<0.00*

<0.00*

Mann-Whitney U post hoc test

p value

<0.00**

0.26

0.03

0.26

††

<0.00**

<0.00**

<0.00**

<0.00**

†††

< 0.00**

< 0.00**

< 0.00**

< 0.00**

Values are presented as mean ± standard deviation

Abbreviations: U-TAP, Urimal Test of Articulation and Phonation; CAS, Childhood apraxia of speech group; FAD, functional articulation disorder; PRES (E), Preschool Receptive-Expressive Language Scale (Expressive); PRES (R), Preschool Receptive-Expressive Language Scale (Receptive); CA, Consonant accuracy

†between the normal group and FAD group, ††between the normal group and CAS group, †††between the FAD and CAS groups

* Statistical significance was accepted for p-values < 0.05, ** Statistical significance was accepted by using Bonferroni method for p<0.017

Table 3. Intragroup comparisons of the mean value of PRES (E) and PRES (R)

PRES(E) (%ile)

PRES(R) (%ile)

p value

Normal group

92.13 ± 8.41

91.62 ± 10.58

0.89

FAD group

64.44 ± 22.33

73.22 ± 24.04

0.34

CAS group

1.37 ± 2.75

17.10 ± 17.45

<0.00*

Values are presented as mean ± standard deviation

Abbreviations: CAS, Childhood apraxia of speech group; FAD, functional articulation disorder; Preschool Receptive-Expressive Language Scale (Expressive); PRES (R), Preschool Receptive-Expressive Language Scale (Receptive)

* Statistical significance was accepted for p-values < 0.05

Results: Percentiles calculated according to age for expressive PRES and receptive PRES score also showed significant intergroup differences. There were significant intergroup differences between normal and CAS, and between FAD and CAS groups except for between normal and FAD group (p > 0.017). The CAS group showed the lowest, 1.37 ± 2.75%ile for expressive PRES and 17.10 ± 17.45%ile for receptive PRES. FAD showed 64.44 ± 22.33%ile and 73.22 ± 24.04%ile for expressive and receptive PRES, respectively. Normal group showed 92.13 ± 8.41%ile and 91.62 ± 10.58%ile for expressive and receptive PRES, respectively (Figure 1). Intragroup comparison of the mean value of expressive PRES and receptive PRES scores showed significantly decrease of expressive PRES score than receptive PRES only in the CAS group (p < 0.05) (Table 3).

Discussion; Our results regarding expressive and receptive PRES scores also tended to increase in the order of CAS, FAD, and the normal control group, revealing the lowest language skills in the CAS group. This is consistent with previous studies, which showed that CAS patients had impaired cognitive language functions. Moreover, our results regarding the mean values of PRES and U-TAP showed similar patterns between the three groups (Fig.1). In 2019, Choi et al. studied the correlation between the values of articulation tests and language tests. Thirty-three children aged 3–6 years were enrolled, revealing that U-TAP and PRES showed a significant correlation [17]. This is consistent with our results, which is another interesting finding of our study. In addition, there was significant difference between the express and receptive PRES scores only in the CAS group, showing significantly decreased result of expressive score than receptive score. This result imply that FAD patients can express what they want to speak even if they have articulation problem, but CAS patients have a significantly lower level of expression skill than the level of their receptive language.

5) “successive U-TAPs”

It would be better to clearly convey that it is a repeated test rather than a series of tests.

 Answer: I appreciate your comments for the manuscript. Following your comments, we revised the manuscript.

Abstract: Four age-matched typically developing children were also recruited. U-TAP was performed twice repeatedly consecutively

Introduction: In the current study, we aimed to evaluate the difference in U-TAP consonant accuracy (CA) by performing U-TAP twice repeatedly in seccession in age-matched patients with CAS, functional articulation disorder (FAD), and normal groups, and to identify the possibility of U-TAP as a screening method for CAS patients.

Assessment: All subjects performed the U-TAP twice repeatedly in succession. CA was measured by dividing the number of correctly pronounced consonants by the total number of pronounced consonants [13]. The mean value and the difference between two repeated consecutive U-TAP CAs were calculated.

Discussion: In the current study, we evaluated the difference in CA using repetitive successive U-TAPs and showed that CAS patients had a significant difference compared to FAD and normal children. The mean value of CA, expressive PRES and receptive PRES scores showed an increase in the order of CAS, FAD, and normal groups. In contrast, the difference between two repetitive successive U-TAP CA showed a decrease in the order of CAS, FAD, and the normal group.

6) To the best of our knowledge, this is the first study to identify the possibility of using 218 repeated U-TAPs as a screening tool for CAS. Do you think a repeated articulation test could be a screening test? As you mentioned in the introduction, there’s no definite quantitative tool to diagnose CAS but only we have diagnostic criteria. I guess repeated tests of U-TAP might be used as a supportive diagnostic tool for CAS rather than a screening.

Answer: I really appreciate your comments for the manuscript. Following your comments, we revised the manuscript.

Abstract: The CAS group showed a significant difference in the two U-TAP CA compared to the FAD and normal groups. This result implies that repeated U-TAP can be useful for supportive diagnostic tool screeningfor CAS by detecting poor reliability of phonation.

Introduction: In the current study, we aimed to evaluate the difference in U-TAP consonant accuracy (CA) by performing U-TAP twice repeatedly in age-matched patients with CAS, functional articulation disorder (FAD), and normal groups, and to identify the possibility of U-TAP as a supportive diagnostic tool screening method for CAS patients.

Discussion: To the best of our knowledge, this is the first study to identify the possibility of using repeated U-TAPs as a supportive diagnostic screening tool for CAS.

Round 2

Reviewer 1 Report

Thank you for the good research.

It has been appropriately modified and has better results. 

Thank you. 

Reviewer 2 Report

The manuscript is well-written after revision. The condition of childhood apraxia of speech is not a common situation, and its diagnosis is not clear-cut. This approach has originality. I appreciate the authors for the creative job.